

# UAV Based In situ Measurements of $CO_2$ and $CH_4$ Fluxes over Complex Natural Ecosystems

Abdullah Bolek[1], Martin Heimann[1,2], and Mathias Göckede[1]

[1]Max Planck Institute for Biogeochemistry, Department of Biogeochemical Signals, Jena, Germany
[2]Institute for Atmospheric and Earth System Research, University of Helsinki, Helsinki, Finland

**Correspondence:** Abdullah Bolek (abolekbgc-jena.mpg.de)

**Abstract.** This study presents an unmanned aerial vehicle (UAV) platform used to resolve horizontal and vertical patterns of $CO_2$ and $CH_4$ mole fractions within the lower part of the atmospheric boundary layer. The obtained data contribute important information for upscaling fluxes from natural ecosystems over heterogeneous terrain, and for constraining hot spots of greenhouse gas (GHG) emissions. This observational tool, therefore, has the potential to complement existing stationary carbon

monitoring networks for GHGs, such as eddy covariance towers and manual flux chambers. The UAV platform is equipped with two gas analyzers for $CO_2$ and $CH_4$ which are connected sequentially. In addition, a 2D anemometer is deployed above the rotor plane to measure environmental parameters including 2D wind speed, air temperature, humidity, and pressure. Laboratory and field tests demonstrate that the platform is capable of providing data with reliable accuracy, with good agreement between the UAV data and tower-based measurements of $CO_2$, $H_2O$, and wind speed. Using interpolated maps of GHG mole

fractions, with this tool we assessed the signal variability over a target area, and identified potential hot spots. Our study shows that the UAV platform provides information about the spatial variability of the lowest part of the boundary layer, which up to this date remains poorly observed, especially in remote areas such as the Arctic. Furthermore, using the profile method, it is demonstrated that the GHG fluxes from a local source can be calculated. Although subject to large uncertainties over the area of interest, the comparison between the eddy covariance method and UAV-based calculations showed acceptable qualitative

agreement.

## 1 Introduction

Quantifying the emissions of greenhouse gases (GHGs) plays a crucial role in understanding the current and future state of global climate change. Among the GHGs, $CO_2$ and $CH_4$ are the two major contributors to climate change, the former due to its abundance in the atmosphere, the latter due to its higher global warming potential (about 28 times more compared to $CO_2$ in a

20 100 year time frame (Andersen et al., 2018, 2023)). The $CO_2$ and $CH_4$ mole fractions in the lower atmospheric boundary-layer may vary significantly due to small-scale variations in surface-atmosphere exchange fluxes and carbon cycle processes caused by heterogeneity in the ecosystem. Manual flux chambers, eddy covariance (EC) towers, aircraft-based measurements, and satellite remote sensing are the conventional tools that are being used to quantify emissions at different scales from sub-meters





to hundreds of kilometers. However, scale separation between these measuring methods is broad, and there is an urgent need

for a method that can bridge the gap between local and regional scales while still being affordable (Bastviken et al., 2022). Manual flux chambers with a typical footprint size smaller than 1 m$^2$ are a commonly used method to measure land surface fluxes (Livingston and Hutchinson, 1995; Goulden and Crill, 1997; Conen and Smith, 1998; Levy et al., 2011). They are easy to operate and applicable for a variety of regions (Conen and Smith, 1998), but resolve only small spatial scales (Baldocchi, 2003), which makes it challenging to upscale measurements for obtaining flux data that is representative for large scales. Over

the past decades, EC flux towers have become one of the most common tools to quantify the carbon fluxes (Baldocchi et al., 2001; Aubinet et al., 1999). Using EC towers, carbon fluxes can be observed continuously, which is essential to understand carbon exchange processes and the impact of environmental controls on their short-term, seasonal or inter-annual variation. Furthermore, areas with the sizes of a hundred meters to several kilometers can be resolved by EC towers (Baldocchi, 2003), depending on measurement heights, wind direction, atmospheric turbulence, and surface characteristics (Chu et al., 2021). The

changing field of view of an EC tower can be approximated with footprint modeling, but inherent uncertainties and location biases may make interpretation of results difficult in complex terrain (Göckede et al., 2008; Chu et al., 2021). Therefore, extrapolating local measurements to larger scales is challenging given the large spatial heterogeneity. Furthermore, EC towers require constant maintenance and power, which might not always be possible, especially in remote places such as the Arctic. As an option for larger-scale flux observations, aircraft-based measurement campaigns can be conducted, addressing the scaling

issues as well as bridging the gaps between bottom-up and top-down estimates (Chang et al., 2014; Sweeney et al., 2015; Parazoo et al., 2016; Wolfe et al., 2018; Barker et al., 2022). However, aircraft-based measurements are expensive, logistically challenging, and have difficulties flying close to the ground.

With recent developments in unmanned aerial vehicle (UAV) and sensor technology, UAVs have become suitable tools to complement existing carbon monitoring networks, address the scale gap issue, and better represent aggregated signals over

heterogeneous landscapes. Compared to alternative approaches, UAVs can provide an ubiquitous, practical, and comparatively inexpensive approach to quantify the variability in surface-atmosphere exchange processes at local to regional scales, whilst particularly addressing the uncertainties associated with upscaling localized information from stationary EC towers in hetero-geneous terrain. UAVs have been demonstrated as reliable tools to measure wind speed and estimate atmospheric turbulence (Neumann and Bartholmai, 2015; Donnell et al., 2018; Palomaki et al., 2017; Shimura et al., 2018; Thielicke et al., 2021; Wetz

et al., 2021; Bolek and Testik, 2022; Wildmann and Wetz, 2022; Wetz et al., 2023) and GHG mole fractions (Gålfalk et al., 2021; Andersen et al., 2018, 2023; Scheller et al., 2022; Morales et al., 2022; Kunz et al., 2018, 2020; Lampert et al., 2020), which both are required to quantify the emission rates. In the past, three different approaches have been applied to quantify the emission rates with UAVs: using a coil-shaped long stainless-steel tubing called Aircore to collect gas samples (Karion et al., 2010; Andersen et al., 2018, 2023; Morales et al., 2022), collecting atmospheric air in discrete samples via flasks (Lampert

et al., 2020), and measuring the in situ mole fractions onboard the UAV with compact GHG analyzers (Gålfalk et al., 2021; Kunz et al., 2018, 2020; Tuzson et al., 2020; Oberle et al., 2019; Liu et al., 2022). Aircore offers great flexibility for UAV-based measurements due to its light weight and ability to provide continuous measurements. However, it is limited in spatial reso-lution (about 40m in horizontal direction) and requires an immediate analysis of sampled air after the flight to avoid the loss



of sample resolution due to molecular diffusion within the sampling tube (Andersen et al., 2018). Sampling with flasks cannot

provide continuous measurements, and the required instrumentation is relatively heavy.

Onboard measurements using compact GHG analyzers can provide continuous measurements with high spatial resolution. Several studies estimated fluxes over landfills using continuous in situ GHG observation with UAVs (Allen et al., 2019; Gålfalk et al., 2021), but so far, only few studies targeting signals over natural terrain have been published. This was mostly due to the low signal-to-noise ratio and/or heavy weights of previously available portable gas analyzers (Shaw et al., 2021), restricting

application to high-flux environments or limiting the total flight time of the UAV platforms. Recently, portable gas analyzers have become more precise and light enough, to be deployed on UAVs with a reasonable flight time ($\sim$20 mins), but to date there are only a few guidelines available for flight strategies that allow to reliably constrain the GHG measurements both qualitatively and quantitatively over natural emission sources/sinks (Scheller et al., 2022; Shaw et al., 2021). Therefore, more studies collecting in situ GHG mole fractions on board a UAV with different flight strategies are needed to improve our

understanding of how to best use these UAV platforms to complement the existing carbon network.

Here, we present a UAV-based monitoring platform instrumented with $CO_2$ and $CH_4$ gas analyzers, and an ultrasonic anemometer to measure 2-D wind speed, air temperature, humidity, and pressure. This setup allows us to quantify $CO_2$ and $CH_4$ mole fractions in the lower atmospheric boundary layer over terrain composed of different landscape features. This provides valuable information to characterize the impact of landscape heterogeneity on GHG patterns in the lower atmosphere and to identify

local emission sources in complex terrains. The aim of this study is to demonstrate the applicability of the developed UAV platform for both qualitatively assessing GHG signal variabilities over heterogeneous landscape and quantifying the GHG mole fractions and fluxes from the lower part of the atmospheric boundary layer. In Sect. 2, the UAV platform and the used methodologies are introduced. Results of the laboratory tests of gas analyzers and different field tests are presented in Sect. 3, and finally, conclusions are given in Sect. 4.

## 2 Field Site Characteristics and Methodologies

### 2.1 Field Site Descriptions

The first field tests of the UAV platform were conducted over the Jena-Experiment field site (50°57'00" N, 11°37'30" E), located north of Jena next to the Saale river in Eastern Germany. The Jena-Experiment has been the home of ongoing biodiversity research since 2002 (Weisser et al., 2017; Roscher et al., 2004). The core area of the experiment consists of several 20m x 20m

vegetation patches and hosts 60 different plant species. The study area for our UAV campaigns at this site is fairly flat, covering approximately 400m x 300m, surrounded by trees along the edges. The average annual precipitation is 587 mm, and the mean annual air temperature is 9.3°C (Roscher et al., 2004).

To test our UAV system in a natural, heterogeneous ecosystem similar to our primary research target (i.e., the Arctic) we have conducted several flights over one of the most heavily investigated areas within the Arctic circle: Stordalen Mire, a subarctic

permafrost peatland in northern Sweden (68°21'N, 19°02'E), underlain by discontinuous permafrost (Bäckstrand et al., 2010) and showing a substantial small scale heterogeneity in terms of soil moisture and vegetation types (Bäckstrand et al., 2010).



The structure of this ecosystem offers a good opportunity to test the UAV platform's capability of detecting the impact of small-scale surface variability on GHG signals in the lower atmosphere. The average annual air temperature and precipitation are -0.6°C and 304 mm, respectively (Malmer et al., 2005). The mire generally experiences two main wind directions (NorthWest - SouthEast) (see Fig. C1) and it is covered by snow between November and April with a maximum depth of 55 cm (Malmer et al., 2005).

## 2.2 UAV Platform Characteristics and Configuration

The UAV platform in our experiments is a hexacopter that can fly for about 20 mins with a scientific payload of 4 kg (PM X6 Pro XL). The rotor-rotor distance is 1.29 m, and the propeller diameter is 55.9 cm. CubePilot Cube Orange is used as a flight controller (Copter 4.4.2), equipped with triple heated internal measurement units (IMUs) and two barometers (https://docs.px4.io/main/en/flight_controller/cubepilot_cube_orange.html). The integrated sensors of flight controller, i.e. gyroscopes and accelerometers, log the motion of the UAV including roll, pitch, yaw, and accelerations in 3-axis which subsequently can be used to correct the wind speed measured by the anemometer. In addition to the GPS unit, a ground-pointing LIDAR is used to increase the vertical stability of the UAV.

The scientific payload of this UAV platform (see Fig. 1a) includes a 2-D anemometer (Trisonica Mini, LI-COR Environment, USA) that measures the wind speed, air temperature, humidity, and pressure. The anemometer is placed 0.65 m above the rotor plane to avoid propeller downwash contaminating the anemometer measurement. This setup was previously found to provide satisfactory results (Palomaki et al., 2017; Shimura et al., 2018; Donnell et al., 2018; Thielicke et al., 2021; Bolek and Testik, 2022). The platform is instrumented with two gas analyzers to measure $CO_2$ (Licor Li-850, LI-COR Environment, USA) and $CH_4$ (Aeris Strato, Aeris Technologies) mole fractions, with both analyzers placed below the rotor planes. The internal pump of the Licor analyzer is removed to connect both analyzers sequentially, and the inlet of the tubing that leads sample air to both units is placed next to the anemometer. A custom-built data logger and power distribution board (see Fig. 1b) log onboard sensor data and provide power for the entire scientific payload. All data is synchronized at a frequency of 2 Hz including secondary GPS data. Selected data (wind speed, wind direction, and GHG mole fractions) can be transferred to the ground control station using a LoRa (Long Range radio communication) board, facilitating real-time data transmission while flying. Table 1 shows the specifications of the scientific payloads deployed on the UAV. The total take-off mass of the platform is 16.6 kg.

**Table 1.** Characteristics of the scientific payload.

| Instrument | Measurement | Sensitivity | Weight (kg) | Power Cons. (W) |
|---|---|---|---|---|
| Aeris-Strato | $CH_4$ | < 1 ppb | 1.8 | 15 |
| Licor Li-850 | $CO_2$ | < 0.1 ppm | 1.2 | 5 |
| Trisonica Mini | 2-D Wind, T, P, Rh | - | 0.05 | 0.32 |





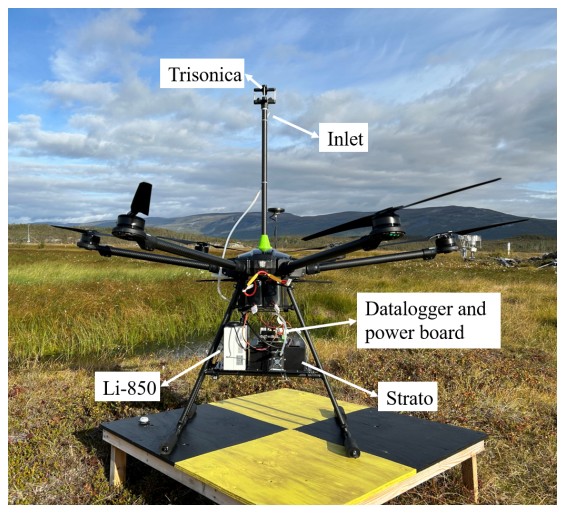

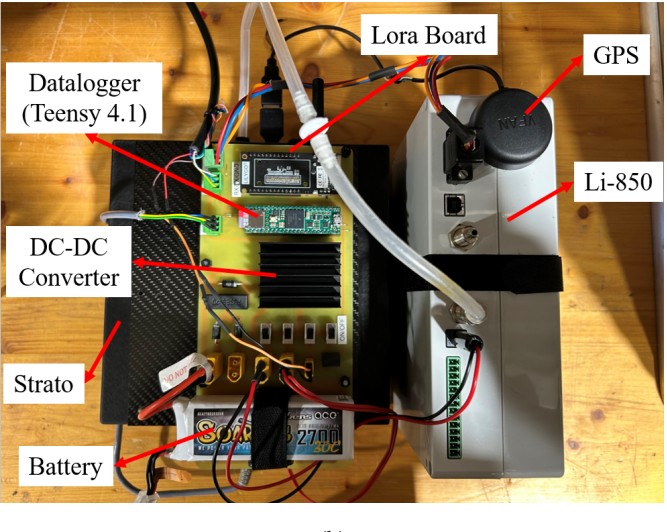

(a)                                          (b)

**Figure 1.** Instrumented UAV platform (a), and close-up picture of the scientific payload (b).

## 2.3 Data Processing

Data collected by the UAV platform was pre-processed to correct or remove low-quality data. The Trisonica mini was set up to
provide 3D wind information; however, the anemometer cannot resolve elevation angles higher than 15°. In addition, vertical
wind speed is prone to biases due to disturbances caused by the propeller downwash. Therefore, we only used 2D wind speed
for further analyses. The wind speed measured by the anemometer on the UAV platform was subjected to disturbances due to
translational and rotational (i.e., roll, pitch, and yaw) motions of the UAV. To compensate for these motions, we followed the
direct correction methods, outlined by Donnell et al. (2018). Here, the heading of the UAV was kept constant during operation
(i.e., no changes in yaw), and the anemometer North was aligned with the UAV's heading. Briefly, we first separated the 3D
wind vector into components (u, v, and w) using wind speed (WS), direction (WD), and elevation angles. Similarly, the UAV's
movement along 3D axes was also calculated from the measured GPS speed, and yaw angles. To compensate the rotational
effects, the perturbations ($r_\theta$, $r_\psi$) due to UAV motions were calculated as follows;

$$r_\theta = \left( \frac{\theta_i - \theta_{i-1}}{\frac{1}{f}} \right) * r \tag{1}$$

$$r_\psi = \left( \frac{\psi_i - \psi_{i-1}}{\frac{1}{f}} \right) * r \tag{2}$$

Here, $\theta$, and $\psi$ represent roll and pitch, respectively, $r$ is the distance between the rotor plane and the anemometer (65 cm),
and $f$ is the sampling frequency of the IMU (Internal Measurement Unit) of the UAV. The true wind speed was obtained





from the raw wind speed ($u_{\mathrm{raw}}$, $v_{\mathrm{raw}}$, and $w_{\mathrm{raw}}$) measured by the anemometer in combination with rotational and translational velocities ($u_{\mathrm{gps}}$, $v_{\mathrm{gps}}$, and $w_{\mathrm{gps}}$) of the UAV as shown in Eqs. 3 and 4 below. It should be noted that opposite sign conventions
were adopted for the velocities of the anemometer and UAV.

$$u = (u_{\mathrm{raw}} + r_{\theta} - u_{\mathrm{gps}}) * \cos(\theta_{\mathrm{i}}) - (w_{\mathrm{raw}} + w_{\mathrm{gps}}) * \sin(\theta_{\mathrm{i}}) \tag{3}$$

$$v = (v_{\mathrm{raw}} + r_{\psi} - v_{\mathrm{gps}}) * \cos(\psi_{\mathrm{i}}) - (w_{\mathrm{raw}} + w_{\mathrm{gps}}) * \sin(\psi_{\mathrm{i}}) \tag{4}$$

Using the yaw angle of the UAV, we aligned the heading of the UAV to true north by rotating the coordinate system:

$$u_{\mathrm{rot}} = u * \cos(\alpha) + v * \sin(\alpha) \tag{5}$$

$$v_{\mathrm{rot}} = -u * \sin(\alpha) + v * \cos(\alpha) \tag{6}$$

Here, $u_{\mathrm{rot}}$ and $v_{\mathrm{rot}}$ are the rotated wind components and $\alpha$ is the difference between the UAV yaw angle and true north. Finally, 2D wind speed (WS) and true wind direction (WD) were calculated as:

$$WS = \sqrt{u_{rot}^2 + v_{rot}^2} \tag{7}$$

$$WD = atan2(v_{rot}, u_{rot}) \tag{8}$$

To remove potential offsets in the calibration of the analyzers (see Section 3.1), we sampled calibration gases with known $CO_2$ and $CH_4$ mole fractions (350 ppm and 550 ppm, and 1700 ppb and 3000 ppb, respectively) before and after each flight day
for about 5 minutes. Note that, only the last 1-2 mins of sampling were used for the calibration process to allow the analyzers cells to be flushed for the first couple of minutes to reach the equilibrium. Observed offsets to the target mole fractions were compensated during the data processing step using simple linear interpolation. Additionally, the $CO_2$ data were subjected to filtering due to observed sporadic spikes during some flights. We first employed hard thresholds that omitted $CO_2$ mole fractions below 380 ppm and above 460 ppm, respectively. In addition, $CO_2$ data were omitted when the absolute difference
between sequential $CO_2$ data was higher than 2 ppm or equal to zero. After eliminating implausible data this way, we applied the despiking algorithm from the RFlux package in R (Vitale et al., 2020) with the default scale parameter and 1-minute window width. The despiking algorithm uses a repeated median filter within the selected window length to find the low-frequency part ($\mu_{\mathrm{t}}$) of the time series and replaces the detected spikes with $\mu_{\mathrm{t}}$ (see (Vitale, 2021) for further details).

## 2.4 Flight Strategies

This study compiles data collected by our UAV platform over the Jena-Experiment and Stordalen Mire, i.e. from a total of 40 independent flights and a combined flight time of about 12 hours. Over these deployments, two distinct flight strategies were tested, namely grid surveys and vertical profile flight.



The grid survey flights were used to qualitatively assess the signal variability over the heterogeneous landscape of Stordalen Mire. Here, our UAV platform was programmed to fly at a constant speed following a pre-defined horizontal grid pattern. The

165 grid surveys started with transects oriented along the east-to-west direction and were subsequently followed by north-to-south legs so that the same locations were sampled twice at two different times at transect intersections. Flight tracks of each survey flight are shown in Fig. 5, with measurement locations marked as black solid circles. The distance between each of these black circles is approximately 2 m based on the sampling frequency of 2 Hz, and the flight speed of 4 m s$^{-1}$.

The vertical profile flights were conducted to observe vertical gradients of atmospheric conditions within the lowest part of

170 the boundary layer and quantify the fluxes using the profile method, described in detail in the next subsection. These flights consisted of multiple waypoints along the z-direction, at which the UAV platform was programmed to hover for 40 or 60 seconds depending on the pre-defined maximum altitude. The vertical resolution of the flight was set to 2.5 m up to a height of 15 m AGL, between 15 to 30 m AGL it was 5 m, and from 30 to 110 m AGL it was 10 m. The starting altitude of the vertical profile flights over the area were set to 5 m AGL (see Fig. 5).

**2.5 Flux Quantification using Profile Method**

This section describes a quantification method (i.e., profile method) that was used to calculate the $CO_2$ and $CH_4$ fluxes based on vertical profile data. In essence, the approach is an application of the flux-gradient method (Xiao et al., 2014; Zhao et al., 2019; You et al., 2021) to UAV-based data. In the flux gradient method, turbulent transport is comparable to molecular diffusion, allowing vertical fluxes to be approximated as the product of the vertical gradient of GHG mole fractions and the eddy

diffusivity (Baldocchi et al., 1988). One of the main advantages of using the profile method in UAV-based calculations is the comparatively short required sampling time, since only mean values of the mole fractions are needed. Firstly, a logarithmic curve was fitted to the vertical mean wind profile as given in Eq. 10 (Foken, 2017; Tagesson, 2012):

$$\overline{WS(z)} = \frac{u_*}{\kappa} \ln \frac{z}{z_0} \tag{9}$$

where $\kappa$ is the von Karman constant [-] that is equal to 0.4, z is the measurement height [m AGL], and $z_0$ is the roughness

length [m]. Eq. 10 can be rewritten as

$$\overline{WS(z)} = a \ln(z) + b \tag{10}$$

where a is the slope of the logarithmic curve fitting defined as $u_*/\kappa$, and b is the intercept. Based on Eq. 10, $u_*$ [m s$^{-1}$] can be estimated using the slope of the logarithmic fittings to the wind speed profiles. Under the assumption of neutral stability and estimation of $u_*$, the eddy diffusivity ($K_{ed}$) can be derived as given in Eq. 11 (Zhao et al., 2019):

$$K_{ed} = \kappa u_* z_g \tag{11}$$





where $z_g$ is the geometric mean of the two altitudes ($z_1$ and $z_2$) between which the flux is being considered. To calculate $K_{ed}$ and the fluxes, we used the lowest two altitudes of our UAV-based vertical profile measurements (i.e., $z_1 = 5$ and $z_2 = 7.5$ m). The reason for this is that near the surface, fluxes are expected to be influenced only by local emissions (i.e. smaller footprint area), while at higher altitudes signals from different sources/sinks over the landscape may affect the measurements (i.e. larger

footprint area), complicating the interpretation of the results. Using $K_{ed}$ the fluxes (F) of $CO_2$ and $CH_4$ can be estimated as given in Eq. 12 (You et al., 2021).

$$F = -K_{ed}\frac{\chi_1 - \chi_2}{z_1 - z_2} \tag{12}$$

$$\chi = \frac{\overline{P}M\overline{C}}{R\overline{T}} \tag{13}$$

In Eq. 13, $\chi_1$, $\chi_2$ are the measured mass concentrations of $CO_2$ [$gCO_2$] or $CH_4$ [$gCH_4$], respectively, at 5 and 7.5 m (Gålfalk

et al., 2021). Here, $\overline{P}$ is averaged pressure [Pa], $\overline{C}$ is measured averaged mole fractions of $CO_2$ or $CH_4$ at each altitude [ppm], M is molar mass [g mol$^{-1}$] with 44.01 for $CO_2$ and 16.03 for $CH_4$, R is the universal gas constant equal to 8.314 [J K$^{-1}$ mol$^{-1}$], and $\overline{T}$ the averaged temperature [K]. The negative sign is introduced as a convention so positive fluxes are emissions from the ecosystem to the atmosphere, and negative fluxes signify uptake. The uncertainty of the flux calculations and friction velocity estimations were performed using Monte Carlo simulations with a probability level of the coverage interval of 0.95

(i.e. Monte Carlo trials of 20000) (Veen and Cox, 2021).

## 3   Results and Discussion

### 3.1   Laboratory Tests of Gas Analyzers

The Licor Li-850 and Aeris Strato gas analyzers used to measure the atmospheric $CO_2$ and $CH_4$ mole fractions were subjected to several tests under a controlled environment. First, a standard air mixture of known mole fractions was sampled by both

analyzers for approximately four hours to quantify the signal stability over time. It was observed that measurements of the Li-850 were subject to a nearly linear drift over time, whereas measurements by the Aeris Strato analyzer partly displayed non-linear fluctuations (see Fig. A1) for around the first two hours. The signal stabilizes two hours after powering up and hence the Strato analyzer was powered up two hours before all the flights that were conducted over Stordalen Mire. In a subsequent step, based on the same four hours time series the noise characteristics of the analyzers were assessed using Allan deviation

plots (Allan, 1987), to specify the optimum averaging time needed to reduce the measurement noise and simultaneously avoid drift contamination (Kunz et al., 2018). The minimum Allan deviation was observed at about 250 s and 32 s for $CO_2$ and $CH_4$, respectively (see Fig. 2). Since the $CH_4$ measurements of the Aeris Strato contained nonlinear drifts, we have used 10 s averaging time for both analyzers. Based on the test results, for 10 s averaging the Allan deviation of $CH_4$ is smaller than 0.25 ppb whereas it is smaller than 0.15 ppm for $CO_2$. To address the issue of non-linear fluctuations in the Aeris Strato signal,





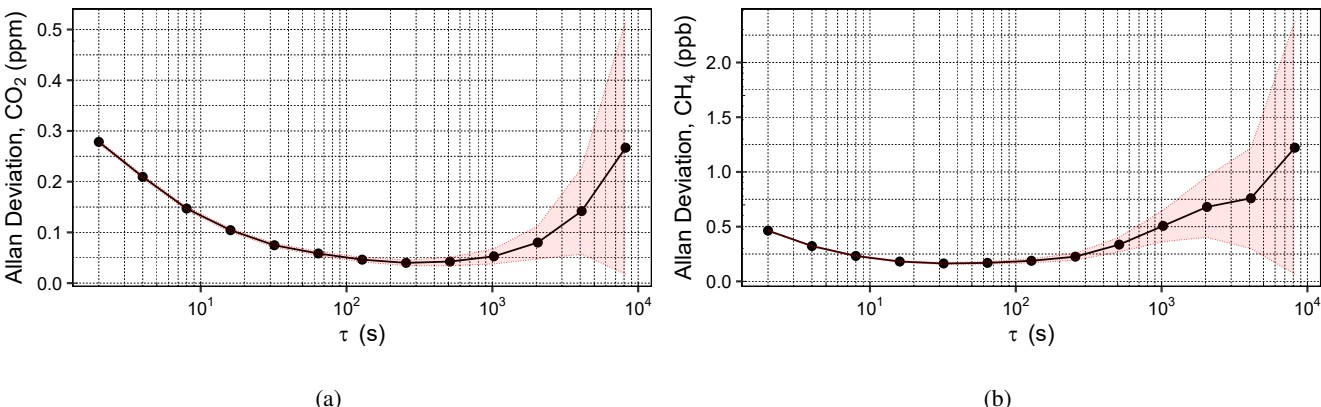

(a)                                    (b)

**Figure 2.** Allan deviation plots of (a) $CO_2$ and (b) $CH_4$. Here, $\tau$ is the sampling time in log-scale and shaded region represents the 95% confidence interval.

we conducted an additional test to quantify the uncertainties of both analyzers using a third gas analyzer (Licor Li-7810) with better temperature stabilization as a reference. For these experiments, both gas analyzers, i.e. the reference LI-7810 and one of the two units used on the UAV, were connected sequentially to a gas tank with known mole fractions of $CH_4$ (3059.21 $\pm$ 0.17 ppb) and $CO_2$ (552.98 $\pm$ 0.02 ppm). The test lasted for an hour, and 10 s averaged root mean square error (RMSE) and mean absolute error (MAE) were calculated as shown in Eqs. 14 and 15:

$$\text{RMSE} = \sqrt{\frac{\sum_{n=1}^{N}(CH_{4,\text{r}} - CH_{4,\text{d}})^2}{N}} \tag{14}$$

$$\text{MAE} = \frac{\sum_{n=1}^{N}(CH_{4,\text{r}} - CH_{4,\text{d}})}{N} \tag{15}$$

Here subscripts r and d denote the reference analyzer (i.e. LI-7810) or UAV (Li-850 for $CO_2$ and Aeris Strato for $CH_4$) after correcting for the mean mole fraction offsets between the analyzers (see Fig. 3). Compared to the reference gas analyzer, the Strato analyzer showed relatively high RMSE and MAE, with the average RMSE at 1.22 ppb and the average MAE of $CH_4$ being almost zero, but with a standard deviation of 1.21 ppb. For the LI-850, the uncertainties in $CO_2$ were found comparable with those of the reference gas analyzer (Fig. 3), with average RMSE of 0.36 ppm and average MAE of $CO_2$ being almost zero, but with a standard deviation of 0.20 ppm. Overall, specified uncertainties in $CH_4$ and $CO_2$ mole fractions were about $\pm$1.2 ppb and $\pm$0.36 ppm, respectively.

## 3.2 Flights over the Jena-Experiment

Over the Jena-Experiment site, we conducted test flights to evaluate the instruments within our scientific payload against tower-based eddy covariance reference measurements. For these test flights, a different calibration gas sampling procedure compared to the one explained in Sect. 2.3 was applied. The field experiment was conducted on 14/07/2023 and the data were compared





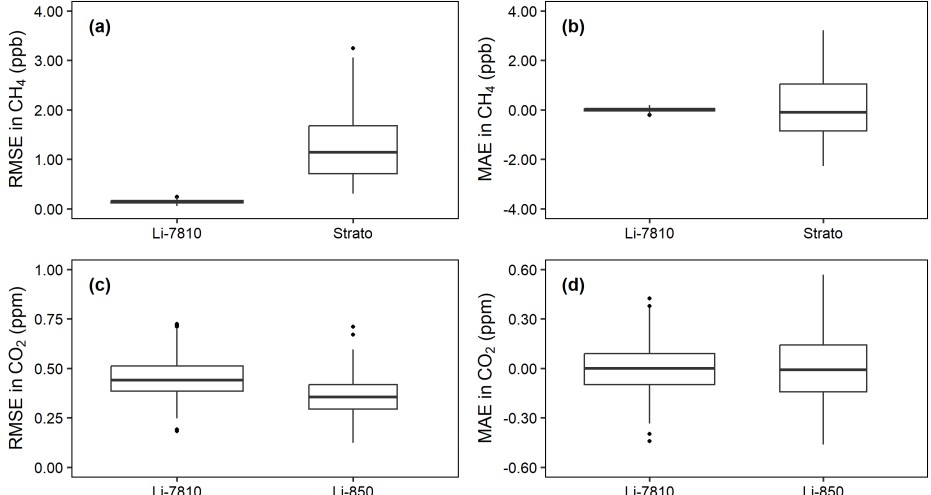

**Figure 3.** 10 s averaged root mean square error (RMSE) and mean absolute error (MAE) of $CH_4$ (a) and (b), and of $CO_2$ (c) and (d), respectively. Here the interquartile range is shown as the rectangle box and the median as the horizontal bar within that box. Solid circles are the potential outliers while the vertical lines represent the minimum and maximum values within the data.

with observations from a 2 m high local tower, which was placed temporarily, instrumented with a 2D anemometer and a Licor
Li-850 analyzer.

Fig. 4 shows the comparison of wind speed, $CO_2$, and $H_2O$ concentrations between UAV and tower datasets. The flight lasted about 21 mins, and the UAV was hovering at 10 m high above the ground level during the entire flight. The reason for not hovering closer to the ground, ideally the same height as the tower, is due to the surrounding trees and fences near the tower location which prevent conducting safe UAV flights at lower altitudes. As there was no additional stationary $CH_4$ analyzer, the
comparison does not include $CH_4$. A calibration gas mixture was sampled from a gas cylinder to quantify the offset between the $CO_2$ analyzers after the flight; no calibrations were performed beforehand. The tank was sampled for 14 mins, and the average offset was corrected for in the subsequent analyses. Overall, good agreement between UAV and tower was observed with a correlation coefficient (r) higher than 0.6 for all measured variables. During the flight, the mean difference between the $CO_2$ concentrations was $3.34 \pm 0.91$ ppm. A large fraction of this difference can be attributed to vertical concentration gradients
within the atmospheric boundary layer (see e.g. reference vertical profiles between 2.5 and 10 m from the meteorological tower of the Lindenberg observatory in Germany in Fig. B1). Another factor influencing the comparison of $CO_2$ signals is that $CO_2$ mixing ratios were not converted to dry-air mole fraction here for both tower- and UAV-based gas analyzers. It should be noted that in the rest of this manuscript, $CO_2$ mixing ratios are reported as dry-air mole fraction. Due to the short measurement time, fluctuations of atmospheric moisture can be neglected, but vertical gradients in $H_2O$ levels may lead to minor absolute offsets
between signals from both analyzers. Furthermore, the fact that the local tower is below the canopy height (due to technical limitations) might also affect the observed gradient. For the wind speed comparison between tower and UAV, we found a high correlation coefficient of 0.639 even though the wind speed correction procedure could not be performed due to the lack of





elevation angle data in the data logger. The highest correlation between tower and UAV (r=0.709) was observed for the $H_2O$ measurements.

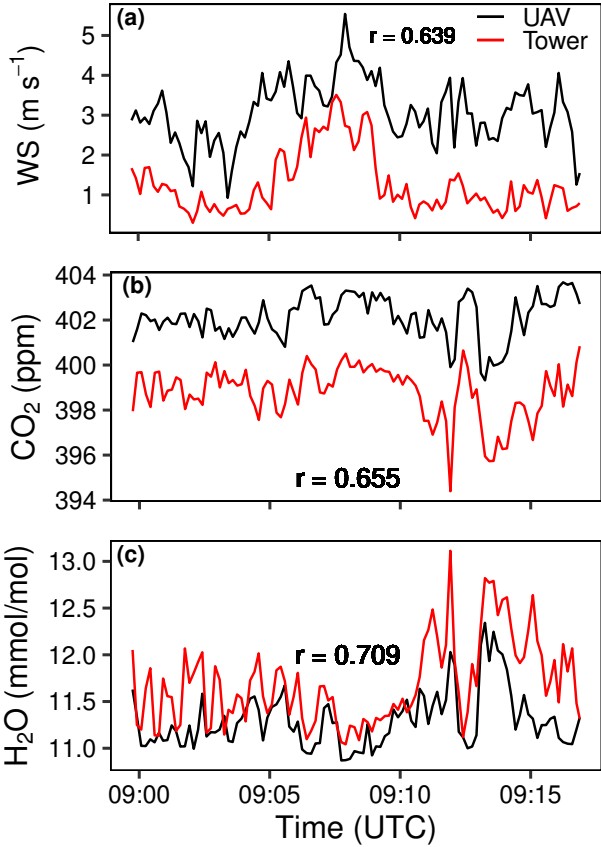

**Figure 4.** 10-s averaged time series of (a) Wind Speed, (b) $CO_2$, and (c) $H_2O$ illustrated by black line for UAV and red line for tower measurements. Here, r represents the correlation coefficient between the tower and UAV.

## 3.3 Grid flights over Stordalen Mire

Several grid survey flights were conducted over Stordalen Mire in the period 11. to 14. of September 2023, with the main focus to identify potential hot spots in atmospheric GHG mole fractions and quantify the signal variations over the areas of interest. Three different target areas were defined using the vegetation map from Varner et al. (2022): two of them, labelled as Area 1 and Area 2 in the following, were classified as bog and fen, respectively, and thus featured mostly wet surfaces. The third domain, Area 3, was mostly dry and was classified as a palsa mire (see Fig. 5). Each of these areas was surveyed two times. All grid survey flights were conducted at an altitude of 10 m AGL, and each flight lasted about 14 minutes. Sampling within each grid survey was interrupted for about 15 to 20 minutes to prepare the UAV platform for the second flight leg. The only exception to this was the second survey over Area 3, for which due to bad weather conditions the second part of the survey



**Table 2.** Flight details over Stordalen Mire.

| Flight Type | Date | Start Time (UTC) | End Time (UTC) | Location | Area ($m^2$) | Flight Speed ($m\,s^{-1}$) | Altitude (m) |
|---|---|---|---|---|---|---|---|
| Vertical Profile | 07/09/2023 | 15:29:09 | 15:42:31 | — | — | 2 | 5 - 50 |
| Vertical Profile | 08/09/2023 | 10:12:10 | 10:25:25 | — | — | 2 | 5 - 50 |
| Vertical Profile | 09/09/2023 | 09:22:13 | 09:34:50 | — | — | 2 | 5 - 110 |
| Vertical Profile | 15/09/2023 | 06:42:57 | 06:50:20 | — | — | 2 | 5 - 50 |
| Grid Survey | 11/09/2023 | 08:19:40 | 08:34:20 | Area 1 | ~15k | 4 | 10 |
| Grid Survey | 11/09/2023 | 08:49:15 | 09:03:25 | Area 2 | ~15k | 4 | 10 |
| Grid Survey | 11/09/2023 | 09:24:08 | 09:37:40 | Area 3 | ~12.5k | 4 | 10 |
| Grid Survey | 13/09/2023 | 07:11:00 | 07:24:59 | Area 1 | ~15k | 4 | 10 |
| Grid Survey | 13/09/2023 | 07:36:30 | 07:50:36 | Area 2 | ~15k | 4 | 10 |
| Grid Survey | 14/09/2023 | 09:13:40 | 09:27:00 | Area 3 | ~12.5k | 4 | 10 |

needed to be postponed to the following day (see also details of the grid survey flights in Table 2). All survey grid data were
subjected to a filtering process to remove observed sporadic spikes in $CO_2$ data as was described in Sec. 2.3.

The mole fractions of $CH_4$ over Areas 1 and 2 were found to be higher than those over Area 3 for both flight days (see Fig. 6).
This can most likely be attributed to higher methane emissions, indirectly corroborated by higher water availability, previously
identified as one of the main drivers of $CH_4$ emissions (Kwon et al., 2022). From the land cover map used within this study
(Fig. 6), we estimated that about 70 - 75% of both Area 1 and Area 2 were either bog or fen, while the fraction of these classes
was estimated at only about 30% for Area 3. Furthermore, we found the variability of the measured $CH_4$ mole fractions to be
highest over Area 3 (average 35.4 ppb), compared to an average 22.9 and 28.5 ppb for Area 1 and Area 2, respectively.

Opposed to our findings, Scheller et al. (2022) found smaller variability of $CH_4$ over dry tundra compared to Fen areas. This
might be due to the different measurement heights of the two studies: here the measurement height was 10 m, while in Scheller
et al. (2022) it was about 0.3 m. Additionally, although Area 3 was specified as mostly dry, it still comprises small patches
of fens and bogs. Since the measurements over Area 3 were conducted on a different date than those for Areas 1 and 2, the
observed variations may also be linked to different environmental conditions; however, the measured $CO_2$ mole fractions were
also higher over Areas 1 and 2, compared to Area 3. Furthermore, the variations of $CO_2$ over Areas 1 and 2 were found about 6
ppm whereas it was about 4.8 ppm over Area 3. Overall, our UAV-based observations over Stordalen Mire demonstrate that the
spatial variability in $CO_2$ and $CH_4$ mole fractions can be high even over small spatial scales. This implies that measurements
made by stationary EC towers may be subject to substantial location biases in complex environments such as Arctic wetlands.

The grid survey data were spatially averaged into 10 grid cells each in latitude and longitude direction, which resulted in a grid
with a spatial resolution of about 15-20 m. Bins with less than 20 data points were excluded from further analysis. We then
interpolated the spatially averaged data using the Ordinary Kriging algorithm (Pereira et al., 2022) to facilitate a visual overview
of the mole fraction distribution over the study areas (see Fig. 7). In this context, best-fitted model variograms were selected
based on the RMSE and coefficient of determination ($R^2$). Using interpolated maps, potential hotspots could be identified.







**Figure 5.** Overview of conducted flights in Stordalen. Flight tracks of grid surveys over three different areas. Here, the landcover map (Varner et al., 2022) shows the vegetation distribution and the border of areas was represented by dashed lines (blue for wet, orange for dry). Flight tracks are indicated with black solid circles. The location of the Vertical profiles was illustrated with a red circle whereas the location of the ICOS tower was depicted with a star.

Respective areas are enclosed by black dashed lines in Figure 7. Potential hotspot areas were identified by first locating the 95 percentile of mole fractions over each flight area and flight day, subsequently the locations with matching hotspots for both survey days were identified on the map. Overall, the gridded datasets showed a range in $CH_4$ mole fractions of up to 25 ppb for a single measurement period across the three target sites, while this range was 6 ppm for $CO_2$ mole fractions. Considering the rather small size of the study domain, this level of variability emphasizes the heterogeneity of ecosystem characteristics and corresponding carbon cycle processes within structured wetlands and highlights the value of survey flight data as presented within this study.





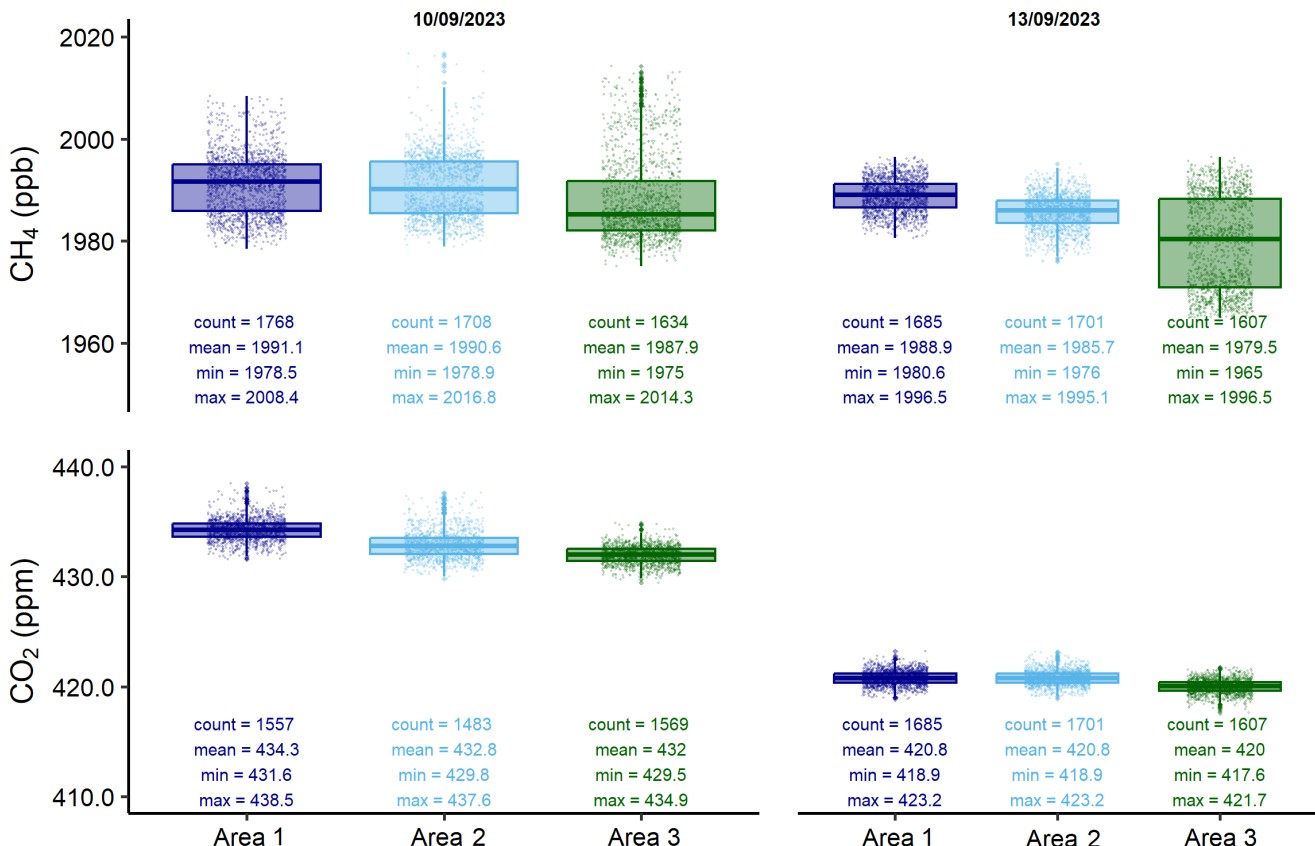

**Figure 6.** $CH_4$ and $CO_2$ mole fractions measured by UAV from the grid survey flights on 11/09 and 13/09/2023. The measured data points (2 Hz) were shown while the outliers were represented as diamonds. Basic statistics including the number of data points as count, mean, min, and max values were denoted underneath each corresponding box plot.

Over Area 3, the eastern part of the domain showed particularly high $CH_4$ mole fractions on both days. These areas are close to Areas 1 and 2, and it is therefore possible that a signal that originated either in Area 1 or Area 2 might be picked up by UAV platform while flying over Area 3 due to horizontal advection. At the same time, enhanced $CH_4$ may also be correlated with the distribution of wet microsites within Area 3. Spatial variability within the $CH_4$ and $CO_2$ mole fraction fields at such small scales, as shown in these horizontal maps, are very challenging to detect with conventional stationary measuring methods. The frequency distribution of observed mole fractions within all three areas is illustrated in Fig. 8. Here, only data from the 11/09/2023 flights were selected, since all areas were sampled within the same day. The means of the mole fractions were 432.99 ppm for $CO_2$ and 1989.93 ppb $CH_4$, with standard deviations of 1.06 ppm and 3.68 ppb, respectively. Figure 8 emphasizes that the mole fractions over the significant section of the total area (about 36 and 27% of $CO_2$ and $CH_4$, respectively) do not overlap with the designated threshold (i.e. $\mu \pm \sigma$, where $\mu$ is the mean, and $\sigma$ is the standard deviation) which





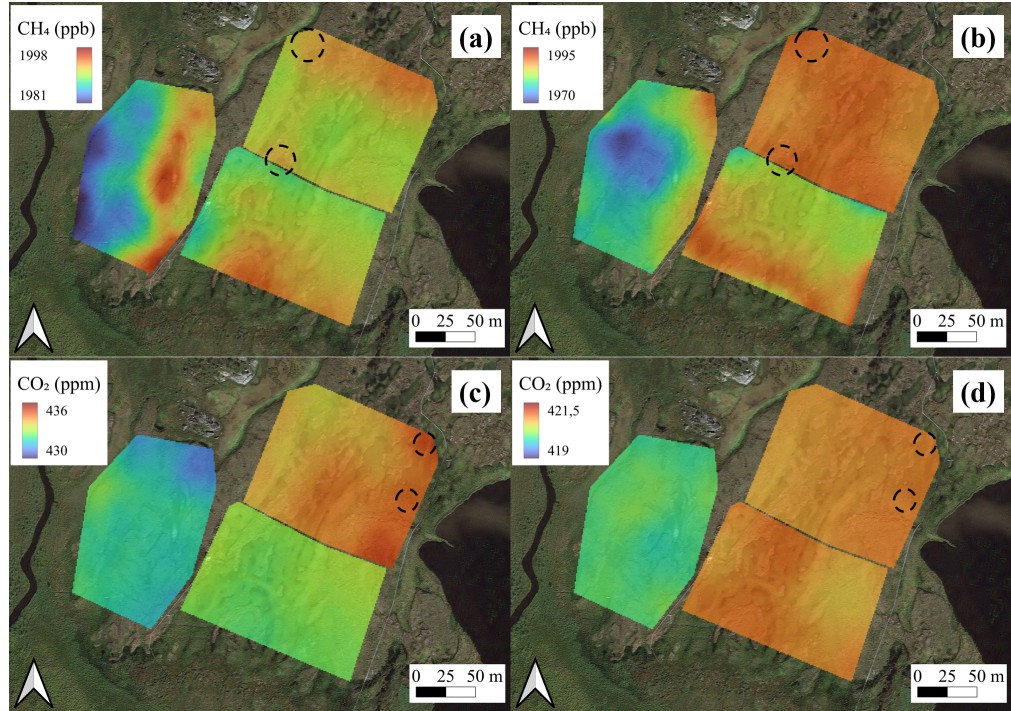

**Figure 7.** Interpolated $CH_4$ and $CO_2$ mole fractions, (overlayed on satellite image from © Google Maps), using Kriging algorithm of the grid surveys that were conducted on 11/09/2023 (a) and (b), and on 13-14/09/2023 (c) and (d). Note that, legends are different for each measurement day to highlight the potential hotspots. Here, color gradients from blue to red were used where blue colors represent low and red colors represent high mole fractions. Potential hotspots were enclosed by black dashed lines.

again highlights the pronounced signal variability over heterogeneous landscapes. Therefore, we recommend combining maps as these with eddy tower and chamber data to improve the interpretation of observational studies over complex ecosystems, and to minimize potential representativeness errors which would e.g. affect modelling frameworks that are trained with such datasets.

### 3.4 Vertical profile flights over Stordalen Mire

Apart from the grid survey flights, vertical profile flights were conducted between 07-09/09 and 15/09/2023, covering elevations of up to 110 m above the ground level (see Table 2). Only the ascending profile was used in our analysis to avoid sampling an air column disturbed by the rotor downwash. The lowest part of the boundary layer profile was created by averaging every 10 s aggregated block at each altitude (see Fig. 9). The two main wind directions (NorthWest - SouthEast) (see Fig. C1) were also captured with the vertical profiles as East-SouthEast (08/09 and 15/09) and West-NorthWest (07/09 and 09/09). $CO_2$ mole fractions show a mostly well-mixed behaviour, except for the profile observed on 08/09 where lower $CO_2$ mole fractions are observed at the mid altitudes, between approximately 15 and 35 meters. Measured vertical profiles presented here are likely affected by an internal boundary layer (IBL), since the area is surrounded by lakes where the surface roughness is small





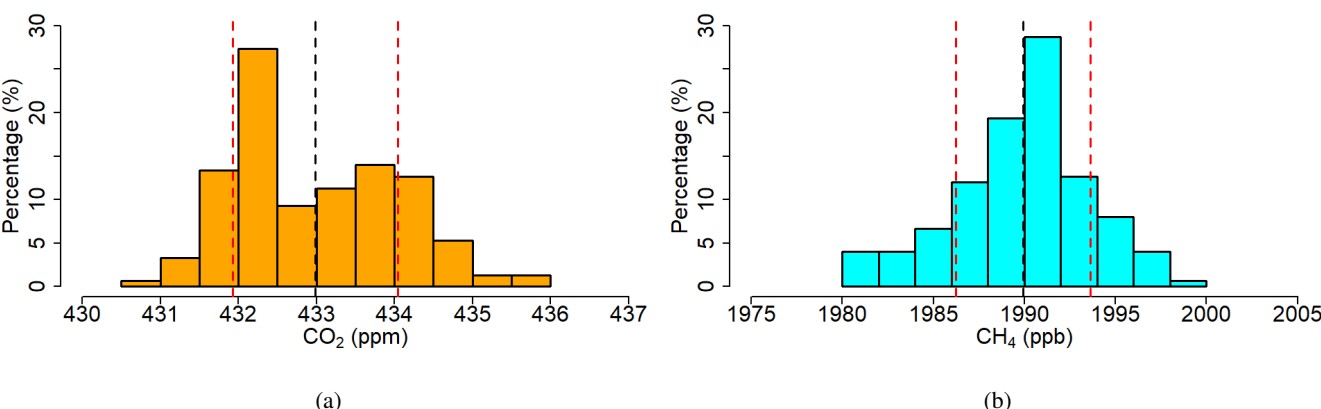

**Figure 8.** The distribution of spatially averaged $CO_2$ (a) and $CH_4$ (b) mole fractions of all three areas combined. Black dashed lines are the corresponding averages while the red dashed lines are the $\pm\sigma$, where $\sigma$ is the standard deviation.

compared to the heterogeneous surfaces around the measurement point, and energy fluxes likely carry different signatures. The formation of the IBL is evident on the profile of 08/09, where the most likely explanation of the observed profile is that the airflow first crossed the entire lake on the eastern side and was subsequently affected by the heterogeneous surface between the lake coast and the measurement location. In our vertical profile, the associated transition layer forms above 15 m AGL, where the wind speed decelerates due to smooth-to-rough transition demonstrated similar to (Krishnamurthy et al., 2023). $CH_4$ mole fractions show high variations for all the profiles and generally are complex to interpret. Nevertheless, higher $CH_4$ enhancement is observed under easterly wind compared to the westerly. Although day-to-day variations might also play an important role, from the landcover map it is clear that the eastern side of the profile measurement point comprises a higher amount of wet areas suspected to be the source of $CH_4$. The lowest $CH_4$ mole fractions were recorded on the 07/09 profile, where the wind direction was $268.8 \pm 21.6°$(i.e., westerly wind) while higher mole fractions were observed when the wind direction was $108.6 \pm 14.4°$and $147.8 \pm 3.1°$(East-SouthEast) on the 08/09 and 15/09 profiles, respectively. The profile on 09/09 is more complex to interpret since the landscape changes back and forth from lake to land on the northwest side of the measurement point. Considering the profile on 08/09 where an IBL formation was inferred, the $CH_4$ profile seems to be also affected by the IBL. The mid altitudes have the highest $CH_4$ mole fractions where the signals might be originating from the area between the measurement point and the lake on the eastern end from the land cover map (see Fig. 5) which is dominated by fen or bogs. $CH_4$ mole fractions decrease at the higher altitudes (above the IBL) where the signal might be forming over the lake.

The quantification method as explained in Sec. 2.5 was applied here for all vertical profiles to constrain $F_{CH_4}$ and $F_{CO_2}$ (see Table 3). All conducted flights show $CH_4$ emissions, and the average emissions when the wind was blowing from the east side of the measurement location ($52.03 \pm 80.68$ mgCH$_4$ m$^{-2}$ d$^{-1}$) was found to be higher compared to those from the westerly directions ($12.82 \pm 40.24$ mgCH$_4$ m$^{-2}$ d$^{-1}$). Higher $CH_4$ fluxes were also observed at the eastern side of the ICOS EC tower, compared to the western side (Łakomiec et al., 2021), supporting our UAV-based observations. On the other hand, except for the





07/09 flight, all the flights show $CO_2$ uptake over the measuring location. Observed $CO_2$ emissions on 07/09 might be due to lower incoming shortwave radiation since the measurement was conducted in the late afternoon (around 17:30 local time). As a

345 rough validation, we used ICOS EC tower data (SE-Sto) to calculate the $F_{CH_4,Twr}$ using the EC method due to the lack of $F_{CH_4}$ data on ICOS portal. However, for the $F_{CO_2,Twr}$ fluxes and $u_{*,Twr}$ data from the ICOS portal was used (Lundin et al., 2024). Here, the calculations were done as close as possible to the sampling times of the vertical profiles using a 30-minute averaging time. For the profiles of 07/09 and 09/09, mean fluxes for a full hour are used, since the flight took place in the middle of two 30-minute datasets. Furthermore, the uncertainties of the tower-based $F_{CH_4}$ were calculated following Mann and Lenschow

(1994). Note that, the $F_{CH_4}$ calculations here may lead to enhanced uncertainties since limited data quality information was available. Still, this preliminary comparison with the EC tower shows acceptable qualitative agreement considering footprint differences between the UAV and ICOS EC tower which are affected by the spatial variability of the fluxes and the difference in measurement heights (i.e. effective measurement height of the UAV-based calculations is $z_g$ which is 6.12 m whereas for ICOS EC tower it is 2.2 m). Although logarithmic fittings that were used to estimate the friction velocities are not perfect

(average $R^2$ values were around 0.31, 0.57, 0.38, and 0.4 for 07/09, 08/09, 09/09, and 15/09, respectively), estimates were in acceptable range since at least 12 altitude levels were used to represent the vertical profiles and the variations can mostly be attributed to the deviations in the stability conditions from the neutral case. The uncertainties of the calculated fluxes are relatively high (see also Table 3). This is mostly due to the small vertical gradient of $CO_2$ and $CH_4$ between the chosen altitudes. In principle, it would be better to select altitudes with a larger mole fraction difference in order to get a better-defined

gradient (Tagesson, 2012), but since the vertical profiles are affected by IBL formation we avoided using higher altitudes. Ideally, setting the lowermost measurement altitude a bit closer to the ground at about 2 - 3 m and the second at around 10 m would be preferable. Also, the measurements were conducted during the late growing season when the fluxes are expected to be small. As a consequence, even though this method for UAV-based quantification of local-scale flux rates is promising, further research will be needed to reduce the uncertainties, e.g. by using different quantification methods and optimized flight

strategies.

**Table 3.** Estimated friction velocities ($u_*$) and fluxes of $CH_4$ ($F_{CH_4}$) and $CO_2$ ($F_{CO_2}$) from the vertical profiles and corresponding uncertainties. Reference values ($_{Twr}$) were derived using observations from the ICOS EC tower at Stordalen Mire.

| Flight Date | $u_*$ (m s$^{-1}$) | $u_{*,Twr}$ (m s$^{-1}$) | $F_{CH_4}$ (mgCH$_4$ m$^{-2}$ d$^{-1}$) | $F_{CH_4,Twr}$ (mgCH$_4$ m$^{-2}$ d$^{-1}$) | $F_{CO_2}$ (gCO$_2$ m$^{-2}$ d$^{-1}$) | $F_{CO_2,Twr}$ (gCO$_2$ m$^{-2}$ d$^{-1}$) |
|---|---|---|---|---|---|---|
| 07/09 | 0.22±0.08 | 0.22 | 9.31±49.56 | 15.84±49.83 | 7.53±9.05 | -1.37 |
| 08/09 | 0.37±0.10 | 0.27 | 34.27±76.29 | 13.95±6.52 | -8.39±17.02 | -3.73 |
| 09/09 | 0.21±0.06 | 0.18 | 16.33±30.91 | -1.36±7.98 | -16.89±10.58 | -6.92 |
| 15/09 | 0.23±0.14 | 0.38 | 69.79±85.07 | 10.86±9.29 | -2.45±10.86 | -3.80 |





**Figure 9.** Vertical profiles of (a) $CO_2$, (b) $CH_4$, (c) Wind speed, (d) Wind direction, and (e) Potential temperatures. Here each symbol represents the average of each 10-s block, and horizontal lines represent the standard deviations. Profiles of Wind speed, $CH_4$, and $CO_2$ close to the surface ($z \leq 10$ m) were provided as a close look-up next to the corresponding figures.





# 4 Conclusions

In this study, we presented a state-of-the-art UAV platform instrumented with in situ $CH_4$ and $CO_2$ gas analyzers and an ultrasonic anemometer capable of measuring 2D wind speed, air temperature, humidity, and pressure. The observational material presented demonstrated how such UAV platforms can be used to collect both qualitative and quantitative information to interpret GHG exchange processes over complex natural terrain.

Two different flight strategies were tested in this study sampling the lowest part of the boundary layer over areas that are otherwise challenging to characterize with stationary devices. Grid survey flights were used to qualitatively represent spatial variability in GHG signals as well as to identify hotspots of the emission sources over a selected study area. Vertical profiles, in turn, were found to be particularly useful for specifying the characteristics within the lower atmospheric boundary layer, filling an important data gap that exists over the Arctic (and other remote areas) primarily due to logistical challenges. Additionally, we have shown that profile flights can be used to quantify the GHG fluxes directly by using the profile method, albeit the data analysed in the scope of this study showed that this approach is subject to large uncertainties and needs further research aimed at their reduction, potentially by employing different flight strategies.

For future studies with the presented UAV platform, combining the grid survey flights with measured wind characteristics via explicit footprint analysis may help to improve the attribution of emission sources. Better localization of the emission hotspots on the surface will allow for improved upscaling of $CH_4$ and $CO_2$, especially if supplemented by stationary EC tower and chamber measurements. In the future the method can also be applied over larger areas, allowing further closure of the scaling gaps e.g. between local observations and satellite remote sensing at regional scales.

In summary, our study shows that UAV platforms are capable of providing valuable information on spatially variable greenhouse gas patterns within the atmospheric boundary layer, which can improve our understanding of greenhouse gas processes within complex landscapes. We have demonstrated the applicability of different flight strategies that can be used to support measurements from an existing carbon flux monitoring network, e.g. to assess signal representativeness for upscaling. However, more flight strategies and quantification methods need to be tested to fully exploit the potential of UAV-based GHG observations for ecological research. This might also be accomplished by conducting synthetic UAV flights within a numerical model.

# Appendix A

The time series of two gas analyzers sampling a calibration mixture are shown in Fig. A1. Here, the linear drift of the Li-850 can be seen, which within 21-mins of flight time the drift is expected to be no more than 0.18 ppm. The Aeris Strato, on the other hand, shows non-linear drifts with a magnitude of no more than 4 ppb over the same period. Apart from the non-linear fluctuations, the drift is usually much less than 4 ppb. These drifts can most likely be attributed to a compromise in temperature stabilization, where the cell enclosure was stripped-down in this instrument model to arrive at a sensor weight compatible with UAV application.

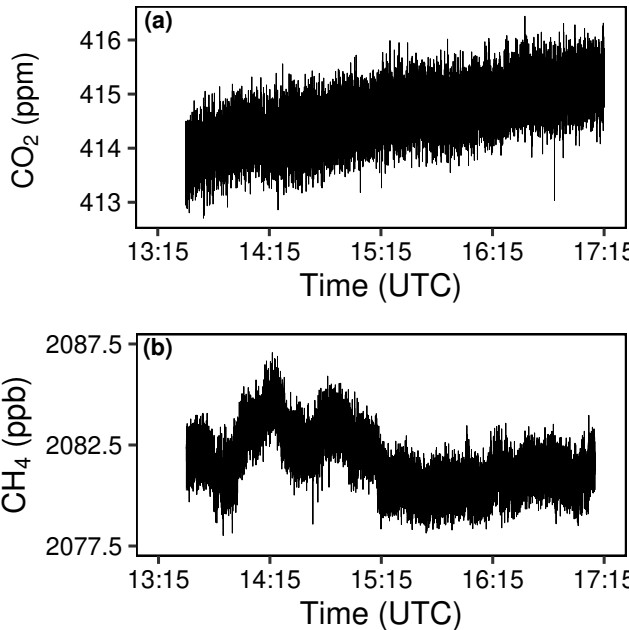

**Figure A1.** Sampling gas analyzers using calibration gas tanks with ambient air mole fractions (a) Licor Li-850, (b) Aeris Strato.

**Appendix B**

A box plot with $CO_2$ concentration differences between measurement heights at 2.5 and 10 m a.g.l. from Lindenberg tower is
shown in Fig. B1. The Lindenberg Tower is part of the Integrated Carbon Observation System (ICOS) network that provides
accurate atmospheric measurements across Europe (ICOS RI et al., 2023). The gradient of the hourly $CO_2$ concentrations
between 2.5 and 10 m above the ground level was found to fluctuate between -1.19 and 1.73 ppm at this site (see Fig. B1).
Here, only summertime measurements (June to August) during the late morning hours (10-12 AM) within the period 2016 to
2022 were considered.

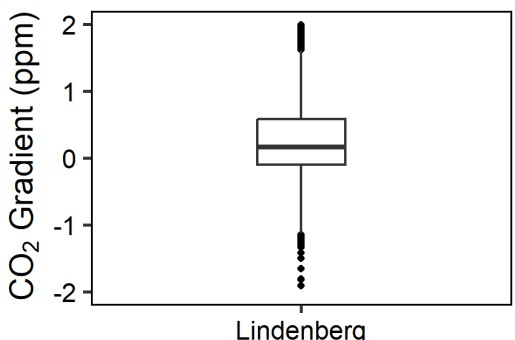

**Figure B1.** Gradient of $CO_2$ calculated from Lindenberg Tower. Two measurement heights of 2.5 and 10 m were used in here.



## Appendix C

Wind characteristics over the Stordalen Mire (ICOS Station ID:SE-Sto) were illustrated using 30-min averaged data between 2021-12-31 and 2023-08-31 (see Fig. C1). This 3-year record shows a clear domination of wind sectors in the WNW and ESE directions, respectively.

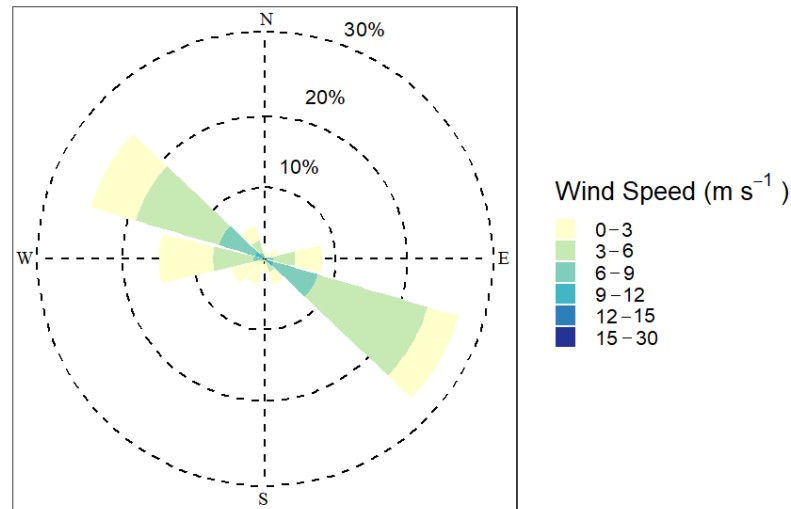

**Figure C1.** The wind rose as measured by the ICOS station located in Stordalen Mire, reflecting mean wind conditions between 2021-12-31 and 2023-08-31.

*Code and data availability.* UAV-based data are available upon request from Mathias Göckede. The data from the Stordalen Mire that was used to create the wind rose can be obtained at ICOS Carbon Portal (https://meta.icos-cp.eu/objects/JFtuqWbso4iTRa0UFYalE-4X, Lundin et al., 2023), together with the Lindenberg tower data for 10 and 2.5 m height levels (https://meta.icos-cp.eu/objects/xif7TvIQNclDN2_eK27_sBl2, https://meta.icos-cp.eu/objects/-5PWpiCSVt4r_49dOTDTdysF, respectively; ICOS RI et al., 2023). UAV-based land cover map data is available at (https://isogenie-db.asc.ohio-state.edu/datasources, Varner et al. (2022)). EC tower data from ICOS Stordalen (SE-Sto) can be found at ICOS Carbon Portal (https://meta.icos-cp.eu/objects/g3HK1QwpR6mug_U-uDedLsTV, Lundin et al., 2024), and $CH_4$ data might be available upon request from ICOS Sweden and the Abisko Scientific Research Station.

*Author contributions.* **Writing/Editing**: Abdullah Bolek, Martin Heimann, Mathias Göckede **Data Collection, Processing, and Analysis**: Abdullah Bolek **Supervision**:Mathias Göckede, Martin Heimann





*Competing interests.* The authors declare that they have no competing interests.

*Acknowledgements.* The presented research was supported by the European Research Council (ERC) under the European Union's Horizon
2020 research and innovation programme (grant agreement No 951288, Q-Arctic). Authors thank the coordinators of Jena-Experiment site for allowing us to conduct the flights, and also ANS researchers for all their support for Stordalen Mire flights. We acknowledge ICOS Sweden and the Abisko Scientific Research Station for providing the eddy-covariance data. ICOS Sweden is funded by the Swedish Research Council as a national research infrastructure. Authors also thank Michał Gałkowski at MPI-BGC/BSI for his valuable comments and suggestions which helped us to improve this manuscript.



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
