# Peer review of "UAV Based In situ Measurements of CO2 and CH4 Fluxes over Complex Natural Ecosystems"

_Atmospheric Measurement Techniques, 2024_

## Referee Comment (RC2)

General comments:

This manuscript presents a novel UAV-GHG platform and its applications on characterizing and quantifying GHG emissions and fluxes for natural ecosystems over heterogeneous terrains. UAV-GHG flux measurement is an innovative topic, and the applied methodology is sound. This paper is well written, and the methodology is clearly presented. It consists of GHG sensors' lab tests including Allan deviation tests. However, how will the sensors perform against temperature changes and water vapor. These parameters would impact the analyzers' performance especially for the field applications. Please refer to Comment 5 and 6 on the laboratory tests. This study conducts demonstration flights in Jena comparing to EC tower measurements and comprehensive grid flights in Stordalen Mire (Arctic ecosystem).

This paper is highly suitable for AMT. I would recommend publication after consideration of the following comments and minor corrections.

Specific comments:

1. Section 3.1 Laboratory tests of gas analyzers would fit better to Section 2.1. Logically, the analyzers should be introduced first before describing the integrated UAV platform. Field site descriptions would be more suitable before the section flight strategies.

2. Section 2.2, how long is the inlet and what are the flow rates for both sensors? Is time synchronization considered for the system (GPS, CO2 and CH4 readings, etc.)?

3. Line 119, what data were pre-processed (from anemometer or GHG sensors)? And how the low-quality data were defined?

4. Line 148 with known CO2 and CH4 mole fractions here, could you track the criterion of these cylinders and provide information here? Please refer to Liu et al., (2022) Laboratory tests part as an example.

5. The long-term test conducted in the laboratory lasted for four hours with a linear drift for CO2. The CO2 sensor may be still warming-up for four hours. Are there any long-term tests over 24 hours performed? Calibration on the field was applied every 24 hours. How large are the sensor's drifts over 24h?

6. Laboratory tests, how was the sensors' performance against water vapor and temperature changes? The field campaign lasts for days, how large is the temperature difference and the humidity during the day? Will these changes during the day impact the sensors' performance?

7. Line 154-155, could you explain the numbers (380 ppm,460 ppm, etc.) chosen to filter the dataset?

8. Table 3 shows the estimated fluxes corresponding to large uncertainties. It would be nice to add a paragraph here to discuss how the large uncertainties were obtained. What are the sources attributed to the uncertainty? Any thoughts to improve the methodology to reduce the uncertainty? The instruments' noise can also impact on the flux error.

Technical corrections:

1. Line 167: Fig.5 shows before Fig.2 in the text. Please correct the order.

2. Line 182: Eq.10 should be replaced by Eq. 9.

3. Line 185: Eq.10 should be replaced by Eq. 9.

References:

Liu, Y., Paris, J.-D., Vrekoussis, M., Antoniou, P., Constantinides, C., Desservettaz, M., Keleshis, C., Laurent, O., Leonidou, A., Philippon, C., Vouterakos, P., Quéhé, P.-Y., Bousquet, P., and Sciare, J.: Improvements of a low-cost $CO_2$ commercial nondispersive near-infrared (NDIR) sensor for unmanned aerial vehicle (UAV) atmospheric mapping applications, Atmos. Meas. Tech., 15, 4431–4442, https://doi.org/10.5194/amt-15-4431-2022, 2022.

---

## Author Response (AR2)

**Response to the comments by reviewer #1**

In the response letter below, we copied the original comments by the reviewer in black, while our answer statements to these comments are printed in blue. All line numbers given in the response refer to the original submission before editing.

This paper describes a study to conduct UAV surveys of GHG profiles (and emissions) using an interesting onboard-UAV GHG analyser in a test study in Jena and then in the Arctic to compare UAV-derived emissions with those from EC towers. The main outputs of the paper relate to the demonstration of the sensor-UAV platform and its uses, and the flux results themselves for e.g. Arctic ecosystems. It would be a valuable and interesting read to those following AMT and a growing community using UAVs for GHG emission work. It is a nice demonstration of a new system.

The paper is well written (thank you for no obvious typos) and well presented. There is careful attention to detail on instrument characterisation and calibration and a clear explanation of the study and its methods (except for flux uncertainties – see specific comments below). I recommend the paper for publication with some thoughts about the relatively minor and constructive comments below.

We thank the reviewer for this very positive overall evaluation of our study.

Specific comments:

A paper by O'Shea et al (below) looked at the spatial scalability of EC and chamber fluxes in the Arctic to 100s km scales using aircraft mass balance. May be useful to briefly discuss this in the intro when discussing Arctic scalability approaches.

O'Shea, S. J. et al.: Methane and carbon dioxide fluxes and their regional scalability for the European Arctic wetlands during the MAMM project in summer 2012, Atmos. Chem. Phys., 14, 13159-13174, doi:10.5194/acp-14-13159-2014, 2014.

**Response to the comment #1**

Thanks for reviewer's suggestion, the manuscript lines 39 – 41 will be revised as follows:

*"As an option for larger-scale flux observations, aircraft-based measurement campaigns can be conducted, addressing the scaling issues as well as bridging the gaps between bottom-up and top-down estimates (O'Shea et al., 2014, Chang et al., 2014; Sweeney et al., 2015; Parazoo et al., 2016; Wolfe et al., 2018; Barker et al., 2022)."*

Line 55: As written, it would indicate that this is an exhaustive list, but it is really only a few examples (so maybe add, "e.g."). A recent paper that has calculated UAV emissions using GHG analysers onboard include the ref below.

Yong, H, et al, 2024: Lessons learned from a UAV survey and methane emissions calculation at a UK landfill, https://doi.org/10.1016/j.wasman.2024.03.025

**Response to the comment #2**

Thanks for reviewer's suggestion, the manuscript lines 52 – 56 will be revised as follows:

*"In the past, three different approaches have been applied to quantify the emission rates with UAVs: using a coil-shaped long stainless-steel tubing called Aircore to collect gas samples (e.g. Karion et al., 2010; Andersen et al., 2018, 2023; Morales et al., 2022), collecting atmospheric air in discrete samples via flasks (e.g. Lampert et al., 2020), and measuring the in situ mole fractions onboard the UAV with compact GHG analyzers (e.g. Galfalk et al., 2021; Kunz et al., 2018, 2020; Tuzson et al., 2020; Oberle et al., 2019; Liu et al., 2022, Yong, H, et al, 2024)."*

Section 2.5 – The flux-gradient method is really interesting. Can you say anything about flux uncertainty here, i.e. can you quantify an uncertainty and what sources of error/bias may affect the fluxes calculated and why? You mention that only a small dataset is needed – this is true for the equations given in themselves, but doesn't a small dataset mean you may not capture any uncertainty or variability? Can you offer more guidance here on the method and its limitations and thoughts on spatial and temporal sampling? I see later that there are +/- flux values in table 3, but it isn't clear how these UAV flux uncertainties have been calculated – are they a statistical variability on many measured fluxes, or are they forward-modelled uncertainties on a single total flux? I see that the uncertainties are sometimes a factor 5 greater than the fluxes themselves (and always >100%) – can you comment on this? There is mention on line 357 that uncertainty is due to the small vertical gradients in GHG concs – but why? To know this, the reader needs info on how flux error is propagated and what it's sensitive to. This needs quite a bit more explanation in the text, as uncertainty is equally (if not more) important than the flux itself (especially when it is higher than the flux itself as it is in this case).

**Response to Comment#3**

Thanks for the reviewer comment. One of the major challenges of the flux-gradient approach is the estimation of the eddy diffusivity parameter. Here we assume a neutral stratification and logarithmic wind profile to first estimate the friction velocity from the mean horizontal wind speed as was described in Foken (2017). To estimate the eddy diffusivity, we used the wind profile method that was described in Zhao et al. (2019).

Another source of the uncertainty is the vertical gradient of the mass concentration of $CO_2$ and $CH_4$ relative to the background signal variability. As the background signal variability increases the uncertainty is expected to increase. Additionally, instrument drift of $CO_2$ and $CH_4$ analyzers are also contributing to the calculated uncertainties in here. However, the differentiation of the background signal variabilities and the instrument drift is not trivial in here since the measurements were conducted over natural ecosystems. The reviewer correctly stated that having small dataset implies less temporal variability is captured, and less information is available for the assessment of uncertainties. This can in principle be compensated having multiple profiles from the same location. However, there is a tradeoff between spatial and temporal variability as the flight time is limited by the battery lifetime.

Accordingly, a decision needs to be made whether sampling multiple profiles over the same location or covering more locations within the target area better serves the objectives of the study. The limitations of the profile method mainly lie in the assumption of neutral stability condition.

Additionally, in some profiles due to the combination of the insufficient sampling time with the non-stationary behavior of the wind speed profiles, the slope of the logarithmic fitting might become negative, in which condition the method cannot be applied. The calculated uncertainty is a modelled uncertainty for a single flux value using Monte Carlo simulation. The Monte Carlo simulations were run using 0.95 of probability which corresponds to 20000 iterations. To do that, we generated synthetic wind speed data assuming normal distribution where we used the measured mean and standard deviation values at each altitude. Similarly, the mass concentrations of $CH_4$ and $CO_2$ were assumed to have normal distribution. Subsequently, using mean and standard deviation of the measured mass concentrations of $CH_4$ and $CO_2$ at each altitude, synthetic mass concentrations data were generated. Using these synthetic data, we estimated uncertainties of friction velocities and calculated the uncertainties for $F_{CO2}$ and $F_{CH4}$. The observed high uncertainties as explained above are mostly related to the combination of high background signal variability and instrument drift relative to the observed vertical gradient of $CH_4$ and $CO_2$ mass concentrations. The result for which an uncertainty almost four times greater than the flux itself was observed might be due to non-stationarity of the observed wind speed profile.

To address the reviewer comment, manuscript lines 205 – 208, 357 – 363, and 379 – 383 will be revised as follows

*"To perform the Monte Carlo simulations, we first generated normally distributed synthetic data for wind speed and mass concentrations of $CH_4$ and $CO_2$ based on the measured means and standard deviations at each altitude. These generated synthetic data were then used to estimate the uncertainties of the friction velocities as well as the fluxes of $CO_2$ ($F_{CO2}$) and $CH_4$ ($F_{CH4}$) (for more details please see Veen and Cox (2021))."*

*"The relatively high uncertainties of the calculated fluxes (see also Table 3) are to the largest part due to the small vertical gradient of $CO_2$ and $CH_4$ relative to the combination of background signal variations and instrument drift. This can be seen e.g. from the panels in Fig. 9 where only the profile part $z \leq 10$ m of the boundary layer was illustrated. In most cases observed in this study, the deviations in the signal are higher than the vertical gradient. Here, the assumption of neutral stability and logarithmic profile might also contribute to the observed high uncertainties. "*

*"As a future profiling strategy, UAV ascending speed will be reduced and the measurements above 25 m AGL will be omitted to avoid the footprint contamination. In addition, the start altitude of the profile flight should be closer to the ground, ideally around 2 – 3 m AGL. This will allow us to have multiple profiles within one flight set, and help to reduce the uncertainties."*

Measuring winds on UAVs: I sympathise with the team and their woes with measuring winds using anemometers on UAVs. It is not easy. There is some recent work on this, where mounting the anemometer more than 2.5 rotor diameter has been shown to negate the flow field problem.

It may be useful to briefly mention that winds remain a challenge but that there are ways to improve (this is also discussed in the Yong et al., 2024 paper referenced above).

**Response to comment #4**

Thanks for the reviewer comment. Indeed, it is a challenge to get the characteristics of the wind speed using an anemometer mounted on a UAV, and the measurements might be biased especially due to propeller downwash. Here, the compromise needs to be made between downwash impact and flight stability, and we think placing the anemometer 65 cm (app. 1.2D) above the rotor plane is a good compromise.

To address the reviewer comment, we will revise manuscript lines 109 – 113 as follows:

*"However, measuring wind characteristics with an anemometer mounted on a UAV still remains a challenge, and compromises need to be made between potential bias due to propellers and the flight stability. We decided to place the anemometer about 1.2D above the rotor plane for best system performance (the potential uncertainty sources and more information can be found in Yong et al, 2024)."*

Technical comments:

Line 84 – space between unit and quantity needed (e.g. "20m") Check throughout.

**Response to technical comment#1**

Thanks for the reviewer comment, we will revise the manuscript line 84 as follows, and checked (and corrected, where needed) similar uses of units throughout the manuscript:

*"The core area of the experiment consists of several 20 m x 20 m vegetation patches and hosts 60 different plant species."*

**References**

Foken, T.: Micrometeorology, pp. 33–81, Springer Berlin Heidelberg, Berlin, Heidelberg, https://doi.org/10.1007/978-3-642-25440-6_2, 2017.

Zhao, J., Zhang, M., Xiao, W., Wang, W., Zhang, Z., Yu, Z., Xiao, Q., Cao, Z., Xu, J., Zhang, X., et al.: An evaluation of the flux-gradient and the eddy covariance method to measure $CH_4$, $CO_2$, and $H_2O$ fluxes from small ponds, Agricultural and Forest Meteorology, 275, 255–264, 2019.

Heimann, M., Jordan, A., Brand, W. A., Lavrič, J. V., Moossen, H., & Rothe, M. (2022). The atmospheric flask sampling program of MPI-BGC, Version 13, 2022. Edmond – Open Research Data Repository of the Max Planck Society, 1-60. doi:10.17617/3.8r.

Liu, Y., Paris, J.-D., Vrekoussis, M., Antoniou, P., Constantinides, C., Desservettaz, M., Keleshis, C., Laurent, O., Leonidou, A., Philippon, C., Vouterakos, P., Quéhé, P.-Y., Bousquet, P., and Sciare, J.: Improvements of a low-cost $CO_2$ commercial nondispersive near-infrared (NDIR) sensor for unmanned aerial vehicle (UAV) atmospheric mapping applications, Atmos. Meas. Tech., 15, 4431–4442, https://doi.org/10.5194/amt-15-4431-2022, 2022.

Yong, H., Allen, G., Mcquilkin, J., Ricketts, H. and Shaw, J.T., 2024. Lessons learned from a UAV survey and methane emissions calculation at a UK landfill. *Waste Management*, *180*, pp.47-54.

O'Shea, S.J., Allen, G., Gallagher, M.W., Bower, K., Illingworth, S.M., Muller, J.B.A., Jones, B.T., Percival, C.J., Bauguitte, S., Cain, M. and Warwick, N., 2014. Methane and carbon dioxide fluxes and their regional scalability for the European Arctic wetlands during the MAMM project in summer 2012. *Atmospheric Chemistry and Physics*, *14*(23), pp.13159-13174.

**Response to the comments by reviewer #2**

In the response letter below, we copied the original comments by the reviewer in black, while our answer statements to these comments are printed in blue. All line numbers given in the response refer to the original submission before editing.

This manuscript presents a novel UAV-GHG platform and its applications on characterizing and quantifying GHG emissions and fluxes for natural ecosystems over heterogeneous terrains. UAV-GHG flux measurement is an innovative topic, and the applied methodology is sound. This paper is well written, and the methodology is clearly presented. It consists of GHG sensors' lab tests including Allan deviation tests. However, how will the sensors perform against temperature changes and water vapor. These parameters would impact the analyzers' performance especially for the field applications. Please refer to Comment 5 and 6 on the laboratory tests. This study conducts demonstration flights in Jena comparing to EC tower measurements and comprehensive grid flights in Stordalen Mire (Arctic ecosystem).

This paper is highly suitable for AMT. I would recommend publication after consideration of the following comments and minor corrections.

We would like to thank reviewer for his/her valuable comments that helped to improve our manuscript.

Specific comments:

1. Section 3.1 Laboratory tests of gas analyzers would fit better to Section 2.1. Logically, the analyzers should be introduced first before describing the integrated UAV platform. Field site descriptions would be more suitable before the section flight strategies.

**Response to Comment #1**

Thanks for the reviewer comment. We will move the section of the laboratory test before describing the UAV platform. The field site description was already given before the section of flight strategies.

2. Section 2.2, how long is the inlet and what are the flow rates for both sensors? Is time synchronization considered for the system (GPS, CO2 and CH4 readings, etc.)?

**Response to Comment #2**

Thanks for the reviewer comment. The inlet tubing is about a 1 m long, and the flow rates of the sensors are about 0.6 l/min. The time synchronization was achieved logging all data to a Teensy microcontroller, here we used Aeris Strato analyzer time, which has an internal RTC (Real Time Clock) to synchronize all the data. However, the Strato analyzer has a problem of deviation from true frequency (i.e. jitter), which means that the collected data is not exactly 2 Hz but about 1.99 Hz. Therefore, we aggregate all the data (UAV and scientific data) to 1s

during the post-processing and do the further calculations. Following the reviewers suggestion, we eliminated the minor time lag between the analyzers and anemometer due to inlet tubing length which was estimated to be around 1 s. We updated the carbon flux calculations and the grid survey flights and the results were slightly changed; however, our main findings have not been affected by this update.

To address the reviewer comment, the manuscript lines 114 – 118, 303 – 308 and 339 – 341, and Figures 6-9 as well as Table 3 will be revised as follows:

*"Due to a frequency deviation issue (i.e., jitter) with the Strato analyzer, the collected data were slightly off from the intended 2 Hz (~1.99 Hz). Therefore, we aggregated all data including UAV movement (translational and rotational motion data), gas analyzers, and anemometer data to 1 s during post-processing step. Additionally, the time lag associated with the inlet tubing length (about 1 s) was also compensated in a post-processing step."*

*"Here, only data from the 11/09/2023 flights were selected, since all areas were sampled within the same day. The means of the mole fractions were 425.12 ppm for $CO_2$ and 2004.62 ppb for $CH_4$, with standard deviations of 1.07 ppm and 3.66 ppb, respectively. Figure 8 emphasizes that the mole fractions over the significant section of the total area (about 35 and 26% of $CO_2$ and $CH_4$, respectively) do not overlap with the designated threshold (i.e. $\mu \pm \sigma$, where $\mu$ is the mean, and $\sigma$ is the standard deviation) which again highlights the pronounced signal variability over heterogeneous landscapes."*

*"All conducted flights show $CH_4$ emissions, and the average emissions when the wind was blowing from the east side of the measurement location ($47.48 \pm 75.13$ $mgCH_4$ $m^{-2}$ $d^{-1}$) was found to be higher compared to those from the westerly directions ($15.62 \pm 39.59$ $mgCH_4$ $m^{-2}$ $d^{-1}$)."*

**Table 3.** Estimated friction velocities ($u_*$) and fluxes of $CH_4$ ($F_{CH_4}$) and $CO_2$ ($F_{CO_2}$) from the vertical profiles and corresponding uncertainties. Reference values ($_{Twr}$) were derived using observations from the ICOS EC tower at Stordalen Mire.

| Flight Date | $u_*$ (m s$^{-1}$) | $u_{*,Twr}$ (m s$^{-1}$) | $F_{CH_4}$ (mgCH$_4$ m$^{-2}$ d$^{-1}$) | $F_{CH_4,Twr}$ (mgCH$_4$ m$^{-2}$ d$^{-1}$) | $F_{CO_2}$ (gCO$_2$ m$^{-2}$ d$^{-1}$) | $F_{CO_2,Twr}$ (gCO$_2$ m$^{-2}$ d$^{-1}$) |
|---|---|---|---|---|---|---|
| 07/09 | 0.22±0.08 | 0.22 | 12.07±47.44 | 15.84±49.83 | 5.23±9.33 | -1.37 |
| 08/09 | 0.37±0.10 | 0.27 | 31.08±69.59 | 13.95±6.52 | -13.11±31.45 | -3.73 |
| 09/09 | 0.21±0.06 | 0.18 | 19.16±31.74 | -1.36±7.98 | -16.58±12.05 | -6.92 |
| 15/09 | 0.23±0.14 | 0.38 | 63.88±80.67 | 10.86±9.29 | -5.29±13.36 | -3.80 |

[Figure]

**Figure 6.** CH$_4$ and CO$_2$ mole fractions measured by UAV from the grid survey flights on 11/09 and 13/09/2023. The measured data points (~2 Hz) were shown while the outliers were represented as diamonds. Basic statistics including the number of data points as count, mean, min, and max values were denoted underneath each corresponding box plot.

[Figure]

**Figure 7.** Interpolated CH$_4$ and CO$_2$ mole fractions, (overlayed on satellite image from © Google Maps), using Kriging algorithm of the grid surveys that were conducted on 11/09/2023 (a) and (b), and on 13-14/09/2023 (c) and (d). Note that, legends are different for each measurement day to highlight the potential hotspots. Here, color gradients from blue to red were used where blue colors represent low and red colors represent high mole fractions. Potential hotspots were enclosed by black dashed lines.

[Figure]

**Figure 8.** The distribution of spatially averaged CO$_2$ (a) and CH$_4$ (b) mole fractions of all three areas combined. Black dashed lines are the corresponding averages while the red dashed lines are the ±σ, where σ is the standard deviation.

[Figure]

**Figure 9.** Vertical profiles of (a) $CO_2$, (b) $CH_4$, (c) Wind speed, (d) Wind direction, and (e) Potential temperatures. Here each symbol represents the average of each 10-s block, and horizontal lines represent the standard deviations. Profiles of Wind speed, $CH_4$, and $CO_2$ close to the surface ($z \leq 10$ m) were provided as a close look-up next to the corresponding figures.

3. Line 119, what data were pre-processed (from anemometer or GHG sensors)? And how the low-quality data were defined?

**Response to Comment #3**

Thanks for the reviewer comment. All data that were collected during flight were pre-processed. Anemometer data were processed to correct the wind speed measurements due to UAV motion (i.e. roll, pitch and yaw). On the other hand, $CO_2$ data were preprocessed due to observed unphysical spikes that were explained in line 153. Additionally, $CH_4$ and $CO_2$ data were processed to correct the possible drift during the flight using calibration gases. Here, the low-quality data were specifically related to the $CO_2$ data where the unphysical spikes were observed. To clarify this in the manuscript, line 119 will be revised as follows:

*"Data collected by the UAV platform was pre-processed to correct or remove low-quality data related to sporadic spikes in $CO_2$ data."*

4. Line 148 with known CO2 and CH4 mole fractions here, could you track the criterion of these cylinders and provide information here? Please refer to Liu et al., (2022) Laboratory tests part as an example.

These cylinders follow the current WMO calibration scales (WMO N2O X2006A, WMO CO2 X2019, WMO CH4 X2014A) through a set of standards that were calibrated by NOAA. More information about these can be found from these flask report https://dx.doi.org/10.17617/3.8r. While working on this reviewer comment, we actually realized the target concentrations of the cylinders were slightly different to those we used previously, which was subsequently fixed. Please see response to Comment #2 for the updated calculations, which show only very minor differences. According to the reviewer request, this information will be added, and the manuscript lines 148 – 153 will be revised as follows:

*"To remove potential offsets in the calibration of the analyzers (see Section 2.1), we sampled high and low, resp., calibration gases with known $CO_2$ and $CH_4$ mole fractions (341.19±0.01 ppm and 543.1±0.01 ppm, and 1722.0±0.1 ppb and 2990.3±0.1 ppb, respectively) before and after each flight day for about 2 to 5 minutes. These gas cylinders were calibrated following WMO calibration scales (WMO CO2 X2019, WMO CH4 X2014A) through a set of standards that were calibrated by NOAA (for more information see Heimann et al. (2022))."*

5. The long-term test conducted in the laboratory lasted for four hours with a linear drift for CO2. The CO2 sensor may be still warming-up for four hours. Are there any long-term tests over 24 hours performed? Calibration on the field was applied every 24 hours. How large are the sensor's drifts over 24h?

**Response to Comment#5**

Our $CO_2$ sensor warms up within 20-30 minutes after powering up. Please see Fig.1 for the cell temperature during the measurements. As can be seen the cell temperature is very stable throughout the measurement period of about 4 hours. Additionally, the calibration on the field was applied each day before and after flights; however, this does not mean that the analyzer kept running for 24 hours. Over the course of one flight (battery allows for about 20 minutes of flight time), the drift is expected to be around 0.18 ppm. Each flight experiment lasted for a maximum of 2-3 hours, and the expected overall drift would thus be about 1-2 ppm. Based on the calibrations executed at start and end of the experiment, this drift will be corrected afterward.

[Figure]

Fig.1. Licor Li-850 cell temperature during 4 hours of measurements.

6. Laboratory tests, how was the sensors' performance against water vapor and temperature changes? The field campaign lasts for days, how large is the temperature difference and the humidity during the day? Will these changes during the day impact the sensors' performance?

**Response to the comment#6**

Unfortunately, we could not conduct tests of sensor performance against changes in the temperature. However, we conduct preliminary test modifying the humidity. We observed that changes of 20 – 25% of relative humidity cause about 1-2 ppm of offset in the $CO_2$ measurements and about 10 ppb in $CH_4$ measurements. However, our measurements were only preliminary since the water vapor measurements needs to be handled carefully, i.e. flushing the analyzer cells and the tubing requires very long time which we were not able to do due to limited resources. Nevertheless, during our flights the observed changes in 10-

s averaged humidity was mostly (~95% of the data) vary within a ±1.5% range. Fig. 2 shows the observed variations of the humidity during all the flights that were conducted in this study. Considering these observed variations in the humidity levels, the offset in measured gas concentrations are expected to be within the provided uncertainty limits. Note that, due to technical issue in anemometer we do not have any humidity data from flight 1009_Area3. To address the reviewer comment, the manuscript lines 233 – 241 will be revised and an Appendix D will be added to the manuscript as follows:

*"In addition to the above-mentioned laboratory tests, we roughly tested the sensors' performance against water vapor and observed that changes in relative humidity of 20-25% caused an offset of about 1-2 ppm in the $CO_2$ and about 10 ppb in the $CH_4$ measurements. Note that, these water vapor tests were preliminary, as accurate water vapor measurements need to be handled carefully, such as by flushing the analyzer cells and the tubing for very long time which was not practiced due to limited resources. Nevertheless, during our flights the observed changes in 10-s averaged relative humidity were relatively small (see Fig. D1). Therefore, the impact of water vapor on the measurements are expected to be minor."*

*"**Appendix D***

*The variations in 10-s averaged relative humidity, after removing mean values of each individual flights, are shown in Fig. D1 for all flights listed in Table 2, except the grid survey flight over Area 3 on 11/09/2023. Due to a technical issue in anemometer, we do not have any humidity data from that flight. Overall, most of the data (about 95%), regardless of flight type, vary within a ±1.5 % range.*

[Figure]

*Fig.2. The variations in 10-s averaged relative humidity for the flights listed in Table 2. Here, mean values of each flights were removed and relative humidity data were grouped based on flight type, i.e. grid survey and vertical profile. "*

7. Line 154-155, could you explain the numbers (380 ppm,460 ppm, etc.) chosen to filter the dataset?

**Response to Comment #7**

The procedure described in the highlighted text passage was intended to eliminate implausible data points captured by the analyzer. To set these hard thresholds, we checked the ICOS tower data of $CO_2$ dry mole fraction between 01/01/2022 – 31/12/2023 (see Fig. 3 below). As can be seen almost all of the data fall within the range 380 to 460 ppm, which is bracketed by red dotted lines.

[Figure]

Fig. 3. Histogram of the $CO_2$ dry mole fraction measured by the ICOS tower for the period 2022/23. Here, the black dashed line represents the mean value, while red dashed lines represent the selected threshold boundaries indicating the plausible data range.

To address the reviewer comment, the manuscript lines between 153 – 154 will be revised as follows:

*"We first employed hard thresholds that omitted $CO_2$ mole fractions below 380 ppm and above 460 ppm, respectively. These plausibility limits were derived from long-term observations of the nearby ICOS tower $CO_2$ measurements."*

8. Table 3 shows the estimated fluxes corresponding to large uncertainties. It would be nice to add a paragraph here to discuss how the large uncertainties were obtained. What are the sources attributed to the uncertainty? Any thoughts to improve the methodology to reduce the uncertainty? The instruments' noise can also impact on the flux error.

**Response to the comment#8**

Reviewer one requested similar additions regarding uncertainties in the flux profile method applied here. Please see our response to the comment #3 of the reviewer #1 for a detailed answer.

Technical corrections:

1. Line 167: Fig.5 shows before Fig.2 in the text. Please correct the order.

**Response to technical comment #1**

Thanks for the reviewer comment. The manuscript line 167 will be removed and line 174 will be revised as follows:

*"The starting altitude of the vertical profile flights over the area were set to 5 m AGL."*

2. Line 182: Eq.10 should be replaced by Eq. 9.

**Response to technical comment #2**

Thanks for the reviewer comment. We will revise the manuscript line 182 as follows:

*"Firstly, a logarithmic curve was fitted to the vertical mean wind profile as given in Eq. 11 (Foken, 2017; Tagesson, 2012):"*

3. Line 185: Eq.10 should be replaced by Eq. 9.

**Response to technical comment #3**

Thanks for the reviewer comment. We will revise the manuscript line 185 as follows:

*"where $\kappa$ is the von Karman constant [-] that is equal to 0.4, $z$ is the measurement height [m AGL], and $z_0$ is the roughness length [m]. Eq. 11 can be rewritten as"*

**References**

Foken, T.: Micrometeorology, pp. 33–81, Springer Berlin Heidelberg, Berlin, Heidelberg, https://doi.org/10.1007/978-3-642-25440-6_2, 2017.

Zhao, J., Zhang, M., Xiao, W., Wang, W., Zhang, Z., Yu, Z., Xiao, Q., Cao, Z., Xu, J., Zhang, X., et al.: An evaluation of the flux-gradient and the eddy covariance method to measure CH4, CO2, and H2O fluxes from small ponds, Agricultural and Forest Meteorology, 275, 255–264, 2019.

Heimann, M., Jordan, A., Brand, W. A., Lavrič, J. V., Moossen, H., & Rothe, M. (2022). The atmospheric flask sampling program of MPI-BGC, Version 13, 2022. Edmond – Open Research Data Repository of the Max Planck Society, 1-60. doi:10.17617/3.8r.

Liu, Y., Paris, J.-D., Vrekoussis, M., Antoniou, P., Constantinides, C., Desservettaz, M., Keleshis, C., Laurent, O., Leonidou, A., Philippon, C., Vouterakos, P., Quéhé, P.-Y., Bousquet, P., and Sciare, J.: Improvements of a low-cost CO2 commercial nondispersive near-infrared (NDIR) sensor for unmanned aerial vehicle (UAV) atmospheric mapping applications, Atmos. Meas. Tech., 15, 4431–4442, https://doi.org/10.5194/amt-15-4431-2022, 2022.